# The Role of the Glutathione System in Stress Adaptation, Morphogenesis and Virulence of Pathogenic Fungi

**DOI:** 10.3390/ijms231810645

**Published:** 2022-09-13

**Authors:** Tanaporn Wangsanut, Monsicha Pongpom

**Affiliations:** Department of Microbiology, Faculty of Medicine, Chiang Mai University, Chiang Mai 50200, Thailand

**Keywords:** glutathione, metabolism, morphogenesis, stress response, virulence, fungal pathogen

## Abstract

Morphogenesis and stress adaptation are key attributes that allow fungal pathogens to thrive and infect human hosts. During infection, many fungal pathogens undergo morphological changes, and this ability is highly linked to virulence. Furthermore, pathogenic fungi have developed multiple antioxidant defenses to cope with the host-derived oxidative stress produced by phagocytes. Glutathione is a major antioxidant that can prevent cellular damage caused by various oxidative stressors. While the role of glutathione in stress detoxification is known, studies of the glutathione system in fungal morphological switching and virulence are lacking. This review explores the role of glutathione metabolism in fungal adaptation to stress, morphogenesis, and virulence. Our comprehensive analysis of the fungal glutathione metabolism reveals that the role of glutathione extends beyond stressful conditions. Collectively, glutathione and glutathione-related proteins are necessary for vitality, cellular development and pathogenesis.

## 1. Introduction

The integration of metabolism with virulence has become a new paradigm in the host–fungal pathogen interaction. In addition to virulence attributes, the term “fitness attributes” has been defined as the cellular functions that are required to support microbial growth and survival [1]. The fitness attributes include the metabolic capacity to assimilate host nutrients, resistance to host-imposed stress, tolerance to elevated host temperature, and the construction of a robust cell wall [2]. The inactivation of fitness attributes will diminish the ability of a fungus to obtain nutrients or combat environmental stressors and hence, will attenuate its ability to grow, express virulence factors, and ultimately cause infection. The metabolism provides a platform for generating the precursors and energy required for growth, antioxidant production and cell wall remodeling. Moreover, metabolic adaptation can modulate the expression of virulence factors and immunogenicity. Thus, the metabolism impacts fungal pathogenicity through both virulence and fitness attributes and is indispensable for a fungal pathogen to colonize and infect a host successfully.

Fungi can infect multiple sites on the human body and cause both superficial and life-threatening infections. As one and a half million people are killed by pathogenic fungi every year, fungal pathogens are known as a “hidden killer” [3]. The most common culprits of human fungal infection come from four major fungal groups: *Candida* species, *Cryptococcus neoformans*, *Aspergillus fumigatus* and thermal dimorphic fungi [4]. Important human fungal pathogens belonging to the thermal dimorphic fungus group are *Talaromyces marneffei*, *Histoplasma capsulatum*, *Coccidioides immitis*, *Paracoccidioides brasiliensis*, *Blastomyces dermatitidis* and *Sporothrix schenckii*. During infection, the host’s innate immune cells, such as macrophages or neutrophils, commonly phagocytize and destroy the fungal cells by generating reactive oxygen species (ROS) and reactive nitrogen species (RNS) [5]. The host macrophages have been reported to produce up to 14 mM hydrogen peroxide and up to 57 uM nitric oxide in response to fungal infections [6,7]. To cope with these host-imposed stressors, fungal pathogens possess antioxidant defense systems with both enzymatic and non-enzymatic mechanisms [8]. In addition, several fungi can switch their morphology to protect themselves from the human immune system. 

Morphological plasticity is one of the main virulence attributes in pathogenic fungi. Cell differentiation and development contribute to diverse morphological changes, including germination, conidiation, morphological switching between yeast and mold forms, and even autolysis. Fungal species generally undergo a morphological transformation during host colonization by responding to specific environmental cues. For example, thermal dimorphic fungi switch morphology from a multicellular mold in environmental niches to a yeast form in warm-blooded hosts due to temperature changes [9,10,11]. Another example is *Candida albicans*, which can be a commensal organism of the human microbiome while also being the most prevalent human fungal pathogen. At least nine distinct cell shapes have already been found in this species. *C. albicans* changes morphotypes when it inhabits different host niches or when it changes between being a commensal or pathogenic organism [12]. In *A*. *fumigatus*, a ubiquitous pathogenic mold, germination of conidia and hyphal growth occurs during an invasive infection of human lungs, while conidiation is strictly inhibited [13,14,15]. The dysregulation of these morphology pathways consistently attenuates the virulence of the pathogenic fungi in animal studies [15,16,17,18]. Overall, cell differentiation and development are critical for fungal morphogenesis and pathogenicity.

This review focuses on the integral role of glutathione metabolism in providing a platform for generating precursors for the antioxidant system and modulating morphogenesis and virulence. Our comprehensive review highlights the role of glutathione metabolism in strengthening virulence and fitness attributes in pathogenic fungi.

## 2. Glutathione Systems

Glutathione (L-γ-glutamylcysteinylglycine) is a crucial metabolite in eukaryotes and plays a major role in protecting cells against oxidative damage [19,20]. Glutathione directly scavenges diverse oxidants, such as superoxide anion, hydroxyl radical, nitric oxide and carbon radicals and is also a cofactor for various antioxidant enzymes, including glutathione peroxidases and glutathione S-transferase [21,22]. There are two states of glutathione in the cells: reduced glutathione (GSH) and oxidized glutathione disulfide (GSSG) (Figure 1). Importantly, GSH is a major tissue antioxidant, while GSSG is accumulated when cells are exposed to increased levels of oxidative stress. Thus, increased ratios of GSSG to GSH are indicative of oxidative stress, and cells tightly maintain levels of reduced glutathione through the balance of its synthesis and reduction.

The GSH/GSSH pathway is composed of five enzymes, which will be the major focus of this review paper. The first two enzymes are involved with glutathione *de novo* biosynthesis via two ATP-dependent steps: the first step is catalyzed by γ-glutamylcysteine synthetase (*GSH1*), while the second step is catalyzed by glutathione synthetase (*GSH2*) (Figure 1). Next, glutathione peroxidase and glutathione S-transferase catalyze the production of GSSH. Glutathione peroxidase is a major enzyme in the defense against hydrogen peroxide (GPx: 2GSH + ROOH → GSSG + H_2_O + ROH, EC 1.11.1.9). Glutathione S-transferase is involved with the detoxification of many xenobiotic compounds by catalyzing the conjugation of substrates to GSH (GST: GSH + RX → GS-X + RH, EC 2.5.1.18), which can then be eliminated from the cells via glutathione conjugate pumps. Lastly, glutathione reductase catalyzes the regeneration of GSH (GSSG + NADPH + H^+^ → 2GSH + NADP^+^, EC 1.6.4.2).

## 3. Glutathione and Role in Oxidative Stress Protection

Molecular mechanisms elucidating how the glutathione system plays a crucial role in the fungal stress response are primarily obtained from model fungi, such as *Saccharomyces cerevisiae*, *Schizosaccharomyces pombe* and *Aspergillus nidulans,* or common human fungal pathogens, such as *C. albicans*, *A. fumigatus* and *C. neoformans*. In thermal dimorphic fungi, however, an understanding of the glutathione system is lacking because glutathione genetic studies have not been extensively explored, as seen in model fungi and other human fungal pathogens; the role of the glutathione system is mostly inferred from gene or protein expression analyses. Therefore, we summarized the experimentally verified functions of each homologous glutathione gene from diverse fungal species (Table 1). This summarized table will be beneficial in predicting the gene functions for other less studied fungi, such as thermal dimorphic fungi or other non-model fungi.

### 3.1. γ-Glutamylcysteine Synthetase and Glutathione Synthetase

As shown in Table 1, the deletion of the genes encoding glutathione synthetic enzymes (*GSH1* or *GSH2* genes or their homologs) resulted in fungal mutants that were glutathione auxotrophs and had an increased sensitivity to various oxidants. Interestingly, the deletion of *gshA* in *A. fumigatus* also impaired cellular iron sensing, indicating crosstalk between glutathione biosynthesis and fungal iron homeostasis [23]. Likewise, the deletion of glutathione synthetase (*GSH2*) from *C. neoformans* resulted in glutathione auxotrophy under iron starvation-induced stress [24]. Indeed, glutathione is proposed to be involved with iron metabolism by its requirement in the Fe–S cluster assembly [25,26]. Importantly, the *GSH1* and *GSH2* genes are essential in diverse fungal species. A study in *H. capsulatum* reported that the deletion of the *GSH1* or *GSH2* genes caused non-viable mutants, indicating that both the *GSH1* and *GSH2* genes are essential. Consistent with the result from *H. capsulatum*, *gshA* (γ-glutamylcysteine synthetase gene) from *Aspergillus oryzae* was also reported to be an essential gene, while the role of the glutathione synthetase gene (*GSH2* homolog) in cellular viability has not been characterized in this fungal species [27]. In *Candida glabrata*, *GSH1*, but not *GSH2*, was an essential gene. These results suggest that the role of the *GSH1* and *GSH2* genes differs from various species in cell viability, and glutathione synthesis has an essential role in iron homeostasis in many fungi.

**Table 1 ijms-23-10645-t001:** Analyses of glutathione-related enzymes in diverse fungal species.

Enzymes in Glutathione System	Species	Gene Name	Phenotypes	References
**γ-glutamylcysteine synthetase**	*Saccharomyces cerevisiae*	*GSH1*	The *gsh1*Δ mutant showed glutathione auxotrophy, slower growth and increased sensitivity to oxidative stress.	[28,29]
	*Schizosaccharomyces pombe*	*gcs1*	- The *gcs1*Δ mutant showed glutathione auxotrophy and sensitivity to cadmium. - The *gcs1*Δ mutant was unable to sporulate.	[30,31]
	*Candida albicans*	*GCS1*	- The *gcs1*Δ mutant showed glutathione auxotrophy, increased ROS production and apoptosis. - The *gcs1*Δ mutant showed no change in morphogenesis and virulence.	[32,33]
	*Nakaseomyces glabrataa (formerly, Candida glabrata)*	*GSH1*	- The *gsh1*Δ mutant was lethal. - A conditional deletion mutant, *gsh1*Δ*pro2-4,* showed low glutathione levels and slower growth in media lacking glutathione. - The *gsh1*Δ*pro2-4* mutant showed sensitivity to oxidative stress (H_2_O_2_, menadione) and cadmium.	[33,34]
	*Histoplasma capsulatum*	*GSH1*	- The *GSH1* gene was expressed only in the yeast form. - The *gsh1*Δ mutant was lethal. - The *GSH1* overexpression mutant showed an inability to switch from yeast to mold form.	[35]
**Glutathione synthetase**	*Saccharomyces cerevisiae*	*GSH2*	The *gsh2*Δ and the *GSH2* overexpression mutants showed normal responses to oxidative stress.	[36]
	*Schizosaccharomyces pombe*	*gsh2*	The *gsh2*Δ mutant showed glutathione auxotrophy and sensitivity to cadmium.	[30,31]
	*Nakaseomyces glabrataa (formerly, Candida glabrata)*	*GSH2*	- The *gsh2*Δ mutant showed glutathione auxotrophy. - The *gsh2*Δ mutant showed low glutathione levels and sensitivity to oxidative stress (H_2_O_2_, menadione) and cadmium. - The *gsh2*Δ mutant showed resistance to tert-butyl hydroperoxide and cumene hydroperoxide stressors.	[34]
	*Cryptococcus neoformans*	*GSH2*	- The *gsh2*Δ mutant showed glutathione auxotrophy under iron starvation conditions. - The *gsh2*Δ showed low glutathione levels and sensitivity to the oxidative stressor diamide, but not H_2_O_2_. - The *gsh2*Δ mutant showed sensitivity to a high salt stressor, the cell wall damaging agent Congo red, and antifungal drugs. - The *gsh2*Δ showed impairment in virulence-related traits, including defects in capsule formation, melanin production and growth at 37 °C.	[24]
	*Histoplasma capsulatum*	*GSH2*	- The *GSH2* gene was highly expressed in the yeast form. - The *GSH2* overexpression mutant showed an inability to switch from yeast to mold form. - The *gsh2*Δ mutant was lethal.	[35]
**Glutathione reductase**	*Saccharomyces cerevisiae*	*GLR1*	The *glr1*Δ mutant showed sensitivity to oxidative stress (H_2_O_2_).	[37,38]
	*Schizosaccharomyces pombe*	*pgr1*	- The *pgr1* overexpression mutant showed resistance to the oxidative stressor menadione but not H_2_O_2_. - The *pgr1* gene expression was induced by various oxidative stressors (menadione, cumeme hydroperoxide and diamide, but not H_2_O_2_), high salt levels (NaCl), high temperatures and starvation. - The *pgr1*Δ mutant was lethal.	[39]
	*Candida albicans*	*GLR1*	- The *glr1*Δ mutant showed sensitivity to oxidative stress (H_2_O_2_) but not formaldehyde or nitrosative stress (NO). - The *glr1*Δ mutant showed an inability to detoxify GSSG. - The *glr1*Δ mutant showed impairment in macrophage killing. - The *glr1*Δ mutant showed decreased virulence, while the *GLR1* overexpression mutant showed increased virulence.	[40]
	*Cryptococcus neoformans*	*GLR1*	- The *glr1* gene expression was induced by nitric oxide (NO). - The *glr1*Δ mutant showed normal morphology. - The *glr1*Δ mutant showed sensitivity to nitric oxide stress but not peroxide stress. - The *glr1*Δ mutant became avirulent in an inhalation model of mouse infection and showed sensitivity to macrophage killing.	[41]
	*Aspergillus nidulans*	*glrA*	- The *glrA*Δ mutant showed slower growth under normal conditions. - The *glrA*Δ mutant showed defects in conidia germination at high temperatures. - The *glrA*Δ mutant showed sensitivity to various oxidants (menadione, diamide and H_2_O_2_). - The *glrA*Δ mutant accumulated a less reduced form of GSH, more intracellular ROS, and had decreased respiration activity.	[42]
	*Paracoccidioides brasiliensis*	*GR*	The vPb18 virulent strain showed increases in both levels of the *GR* gene and enzymatic activity.	[43]
**Glutathione peroxidase**	*Saccharomyces cerevisiae*	*GPX1-3*	- *GPX1-3* genes encoded for phospholipid hydroperoxide glutathione peroxidase. - The *GPX3* product was a major glutathione peroxidase. - The *GPX3* gene was constitutively expressed. - The *GPX1* gene expression was induced under glucose starvation. - The *GPX2* gene expression was induced by many oxidative stressors. - The *gpx3*Δ mutant showed sensitivity to peroxides (H_2_O_2_ and tert-butyl hydroperoxide). - The *gpx1*Δ and *gpx2*Δ mutants showed no sensitivity to oxidative stress. - The *gpx1*Δ*gpx2*Δ*gpx3*Δ mutant showed sensitivity to H_2_O_2_ and phospholipid hydroperoxide (polysaturated fatty acid linolenate 18:3).	[44,45]
	*Candida albicans*	*GPX3* *(ScGPX1 homolog)*	- The *gpx3*Δ mutant (*orf19.4436*Δ) showed sensitivity to H_2_O_2_ and was defective in hyphal formation within macrophage cells. - The *gpx3*Δ mutant showed impairment in killing macrophages and *Galleria mellonella*. - The *gpx3*Δ mutant showed normal virulence in a murine model of infection.	[46,47]
		*GPX31-* *33* *(ScGPX3/* *HYR1 homolog*	- The *GPX31* is a major glutathione peroxidase. - T*he gpx31*Δ (*orf19.86*Δ) and the *gpx31*Δ*gpx32*Δ*gpx33*Δ mutant (*orf19.86*Δ*orf19.85*Δ*orf19.87*Δ) showed sensitivity to oxidative stressors (H_2_O_2_ and t-butylhydroperoxide but not menadione), UV light, heavy metals (cadmium and silver), and cell wall stressors (congo red and calcofluor white).	
	*Cryptococcus neoformans*	*GPX1*, *GPX2*	- *GPX1* and *GPX2* gene expressions were induced in response to t-butylhydroperoxide and cumene hydroperoxide and repressed in response to nitric oxide. - *GPX2* gene expression was induced in response to the hydrogen peroxide stressor. - The *gpx1*Δ and *gpx2*Δ mutants showed normal morphology, melanin production and capsule formation. - The *gpx1*Δ and *gpx2*Δ mutants showed sensitivity to cumene (hydroperoxide) but not superoxide, hydrogen peroxide or nitric oxide stressors. - The *gpx2*Δ mutant showed higher sensitivity to cumene hydroperoxide than the *gpx1*Δ mutant at high concentrations. - The *gpx1*Δ mutant, but not the *gpx2*Δ mutant, showed sensitivity to the peroxide stressor t-butylhydroperoxide. - The *gpx1*Δ and *gpx2*Δ mutants showed sensitivity to macrophage killing, yet the mutants were still virulent in a mouse model.	[48]
	*Aspergillus fumigatus*	*hyr1 (ScGPX3/* *HYR1 homolog)*	- *hyr1* gene expression was upregulated in hyphae and conidia when exposed to neutrophils. - The *hyr1* gene expression was induced when exposed to H_2_O_2_.	[49,50]
	*Talaromyces marneffei*	*gpx1 (ScGPX3/HYR1 homolog)*	- Gpx1 is an antigenic protein. - *gpx1* gene expression was upregulated in the yeast form.	[51,52]
**Glutathione *S*-transferase**	*Saccharomyces cerevisiae*	*GTT1-2*	- *GTT1* gene expression was induced during the diauxic shift and stationary phase. - The *gtt1*Δ, *gtt2*Δ, and *gtt1*Δ*gtt2*Δ showed sensitivity to heat shock in a stationary phase and slower growth at a high temperature of 39 °C. - The *grx1*Δ*grx2*Δ*gtt1*Δ*gtt2*Δ mutant showed sensitivity to xenobiotics (1-chloro-2,4-dinitrobenzene), heat and the oxidative stressors (cumene hydroperoxide and H_2_O_2_).	[53,54]
	*Schizosaccharomyces pombe*	*gst1-3*	- The *gst1*Δ*gst2*Δ and *gst3*Δ mutants showed sensitivity to peroxide stressors (H_2_O_2_ and t-butylhydroperoxide) and the antifungal drug fluconazole. - The *gst1*Δ*gst2*Δ and *gst3*Δ mutants showed resistance to the peroxide stressor diamide. - *gst1, gst2,* and *gst3* gene expressions were induced during the stationary phase and in response to hydrogen peroxide. - All Gst1, 2 and 3 enzymes have glutathione transferase activity, and the Gst3 enzyme also has glutathione peroxidase activity.	[55]
	*Candida albicans*	*GST2*	- The *gst2*Δ mutant showed sensitivity to oxidative stress (H_2_O_2_). - *GST2* gene expression was induced under nitrogen limitation. - The *gst2*Δ mutant showed defects in hyphal switching under nitrogen starvation-induced filamentous growth.	[56]
	*Aspergillus nidulans*	*gstA*	- The *gstA*Δ mutant showed sensitivity to the oxidant diamide, the fungicide carboxin, various xenobiotics (pyrrolnitrin and sulphanilamide), and heavy metals (selenium, silver and nickel). - The *gstA*Δ mutant showed normal growth in the presence of 1-chloro-2,4-dinitrobenzene. - The *gstA* gene expression was induced in response to xenobiotics (1-chloro-2,4-dinitrobenzene) and oxidative stress (H_2_O_2_).	[57]
	*Aspergillus fumigatus*	*gstA*, *gstB*, *gstC*	- All *gstA, B and C* enzymes have both glutathione transferase and glutathione peroxidase activities. - The *gstA* and *gstC* genes were constitutively expressed under normal conditions, and their expression levels were inducible in response to oxidative stress (H_2_O_2_). - The expression of all *gst* genes was induced in response to xenobiotics (1-chloro-2,4-dinitrobenzene).	[58]
	*Paracoccidioides brasiliensis*	*GST1-3*	The vPb18 virulent strain showed increased levels of the *GST1-3* genes.	[43]
	*Paracoccidioides lutzii*	*GST*	GST was exclusively secreted in the yeast form.	[59]

### 3.2. Glutathione Reductase

In addition, glutathione reductase is required for resistance to oxidative stress because the deletion of the glutathione reductase gene commonly results in fungal mutants that are sensitive to various stressors. The details of the growth and stress response defects in deletion mutants are different according to the species. For example, in *S. pombe* yeast, glutathione reductase is indispensable for growth, as the *pgr1*Δ strain was not viable due to the accumulation of GSSG [39,60]. In *S. cerevisiae* and *C. albicans*, the *grl1*Δ mutant was viable and only showed growth defects under oxidative stress [37,38,40]. In *C. neoformans*, the *gr1*Δ mutant grew normally under normal conditions and was sensitive to only nitric oxide stress but not to peroxide stress [41]. In *A. nidulans*, the *grlA*Δ mutant exhibited growth defects, even under normal conditions, yet was still viable [42,61]. The *grlA*Δ strain of *A. nidulans* was also defective in its growth under high temperatures. Overall, these results suggest that glutathione reductase functions differently among fungal species.

### 3.3. Glutathione Peroxidase

Furthermore, glutathione peroxidase plays a crucial role in protecting fungi against oxidative stress since the absence of glutathione peroxidase gene(s) results in fungal strains that are not able to cope with various oxidants, especially peroxides. Nonetheless, distinct cellular responses to oxidative stress could be observed among fungal species. For example, the *gpx3*Δ mutants from *S. cerevisiae* and *C. albicans* were highly sensitive to H_2_O_2_, while the *gpx1*Δ and *gpx2*Δ mutants from *C. neoformans* were not sensitive to H_2_O_2_ but were sensitive to other peroxides (Table 1). Furthermore, the *gpx1*Δ and *gpx2*Δ mutants from *S. cerevisiae* and *C. albicans* did not show any defects in response to oxidative stress, and hence, *GPX3* is proposed to be the main gene encoding for glutathione peroxidase in these fungi. In *T. marneffei*, the glutathione peroxidase gene (*gpx1*; the homolog of the glutathione peroxidase *HYR1* gene from *S. cerevisiae*) was isolated as one of the antigenic proteins [51]. The expression levels of the *T. marneffei gpx1* gene were high in the pathogenic yeast form and were relatively unchanged in the conidia or mold forms. These results imply that glutathione peroxidase contributes to immunological response during *T. marneffei* infection and plays an important role in the pathogenic yeast phase. Collectively, these results suggest that glutathione peroxidase is required for the general oxidative stress defense mechanisms yet could respond distinctly to the different stressors, depending on the fungal species.

### 3.4. Glutathione S-Transferase

Glutathione S-transferase is involved in the resistance to xenobiotics because this enzyme can detoxify a broad range of harmful substances. In addition, glutathione S-transferase can also protect the cells against oxidative stress as it possesses GSH-dependent peroxidase activity. Accordingly, the glutathione S-transferase gene deletion mutants from a wide range of fungi became sensitive to both xenobiotics and various stressors. As seen in the case of other glutathione-related genes, there are differences in the glutathione S-transferase function among individual fungal species. In *S. pombe*, the *gst1*Δ*gst2*Δ and *gst3*Δ mutants were sensitive to the antifungal drug fluconazole, suggesting the role of glutathione S-transferase in mediating drug resistance. In *C. albicans*, the *GST2* gene was additionally induced under nitrogen starvation [56]. In *S. cerevisiae*, the *gtt1*Δ, *gtt2*Δ, and *gtt1*Δ*gtt2*Δ mutants showed an increased sensitivity to heat shock or exhibited growth defects at high temperatures. In *A. nidulans*, the *gstA*Δ mutant was sensitive to heavy metals [57]. Taken together, glutathione S-transferase is an important enzyme that protects fungal cells from diverse types of substances and stressors.

## 4. The Role of Glutathione in Fungal Morphology, Cellular Development and Virulence

Morphological transformation is an important developmental process in fungi. In the case of fungal pathogens, the ability to produce conidia, germinate, and switch morphology is highly linked to virulence. Multiple signaling pathways and transcription factors tightly regulate fungal morphogenesis, and the details have been extensively discussed elsewhere [4,12,16]. However, less is known about how the glutathione pathway specifically contributes to fungal morphogenesis, cellular development, and virulence. Importantly, genetic studies (Table 1) have demonstrated that several mutants defective in glutathione pathways are not only sensitive to oxidative stress but also exhibit abnormal cellular development. In the thermal dimorphic fungi, *H. capsulatum*, the *GSH1* and *GSH2* genes were highly expressed in the pathogenic yeast form, and the *GSH1* and *GSH2* overexpression strains were unable to switch to the mold form [35]. Consistent with this result, the *gpx1* gene from *T. marneffei* was specifically expressed in the pathogenic yeast form [51,52]. Likewise, the *gst* gene from *Paracoccidioides lutzii* was exclusively secreted in yeast but not in mycelial cells [59]. Overall, these results demonstrate that glutathione genes are highly expressed in the pathogenic yeast form of thermal dimorphic fungi, suggesting that the glutathione pathway is important during the morphological transition and infection.

The glutathione system contributes to the morphological and cellular developments, not only in thermal dimorphic fungi but also in other fungal species. In contrast to the thermal dimorphic fungi that switch to yeast during infection, *C. albicans* switches from yeast to filamentous growth in order to penetrate tissue during host infection. Accordingly, the *C. albicans gpx3*Δ mutant was defective in hyphal formation within macrophage cells, and the *gst2*Δ mutant displayed hyphal switching defects under nitrogen starvation-induced filamentous growth [46,56]. In the filamentous fungus *A. nidulans*, the *glrA*Δ mutant showed conidia germination defects at high temperatures, even under non-oxidative stress conditions [61]. In *A. fumigatus*, the transcript of the *gpx3* gene is highly induced in hyphae and conidia upon exposure to human neutrophils [49]. In the pathogenic yeast, *C. neoformans*, the ability to form a capsule, produce melanin and grow at a high temperature are required for this fungal pathogen to establish infection. The *C. neoformans gsh2*Δ mutant, however, was unable to form a capsule or produce melanin [24]. The mutant also had growth defects at 37 °C [24]. In the non-pathogenic yeasts, *S. cerevisiae* and *S. pombe*, the *gsh1*Δ mutant was defective in sporulation, an important developmental process in response to nutrient starvation [31,62,63]. To conclude, the glutathione pathway is necessary for morphological and cellular development in a myriad of fungal species.

Given the central role of the glutathione system in detoxification, cellular development and the oxidative stress response, one may speculate that the glutathione system will impact the ability of fungal pathogens to invade host tissue and cause disease in humans. Based on the data collected here (Table 1), the pathogenic fungi lacking glutathione components commonly became avirulent or had attenuated virulence in macrophage-killing models and/or animal models of infections. In *C. albicans*, the *gpx3*Δ mutant showed impaired virulence in macrophage killing and *Galleria mellonella* infection models [46]. Even though the morphology of the *C. albicans gcs1*Δ mutant was unaffected, this strain was avirulent in mice [32,33]. Likewise, the *glr1*Δ mutant was less virulent, and *GLR1* overexpression showed increased virulence in *C. albicans* [40]. In *C. neoformans*, the *glr1*Δ mutant exhibited normal morphology; however, the mutant was more sensitive to macrophage killing and was avirulent in an inhalation model of mouse infection [41]. Furthermore, the *C. neoformans gpx1*Δ and *gpx2*Δ were more sensitive to macrophage killing yet were still virulent in a mouse model and displayed normal morphology [48]. In the thermal dimorphic fungi, *Paracoccidioides brasiliensis*, the gene expression and enzymatic activity of glutathione reductase demonstrated increased levels in the virulent strain [43]. Similarly, the expression of the *GST1*, *GST2* and *GST3* genes was upregulated in the virulent strain of *P. brasiliensis* [43]. These results suggest that the glutathione pathway is implicated in virulence in *Paracoccidioides* spp. To date, however, none of the glutathione genes from thermal dimorphic fungi have been experimentally validated in regards to their role in virulence during infection. In summary, the generation of glutathione mutants, together with utilizing animal models of infection, have experimentally verified the critical role of glutathione in fungal virulence.

## 5. Glutathione as Modulators of Fungal Virulence and Pathogenesis

As discussed comprehensively in this review, the glutathione system plays a pivotal role in fungal species. Mutants lacking components of the glutathione system demonstrate deleterious changes in the phenotypes associated with fitness and virulence. Glutathione, however, has not oftentimes been thought of as a “regulatory signal” used by pathogens to modulate virulence traits. Recently, several reports support the role of glutathione in regulating virulence in diverse bacterial pathogens [64]. For instance, *Listeria monocytogenes*, a causative agent of listeriosis, uses both host-derived and endogenous glutathione as a critical signaling molecule to regulate the switch from saprophytic to pathogenic lifestyles. In particular, host-derived glutathione triggered increased bacterial glutathione levels. Then, the elevated bacterial glutathione allosterically bonded to the master virulence regulator PrfA, activating the expression of the virulence genes in this pathogen [65]. In *Pseudomonas aeruginosa*, an opportunistic bacterial pathogen, the bacterial-derived glutathione directly activated the global transcription factor Vfr through a reduction reaction [66]. Once Vfr is in a reduced and activated form, Vfr can regulate the expression of protein-secretion systems, which are essential for establishing infection in this bacterial pathogen. Thus, bacterial pathogens can sense glutathione levels and use glutathione to allosterically bind or reduce the disulfide bonds of a transcription factor to regulate virulence genes directly. These examples clearly demonstrate that glutathione is a critical signaling molecule that activates the virulence pathways. Whether or not fungal pathogens can use glutathione as a modulator of their transcriptional regulation to directly control virulence pathways has yet to be explored.

Several studies have provided clues that glutathione may function as a modulator of morphogenesis and virulence in fungi (Table 2). It has been previously hypothesized that intracellular glutathione levels may function as a signaling molecule because redox imbalances, intracellular ROS accumulation and antioxidant enzyme activation can initiate the cell differentiation processes, such as germination, conidiation and dimorphic transition [67]. The first evidence came from studies showing that depletion of intracellular glutathione levels during yeast to hyphal transition was commonly observed in *C. albicans* and *A. pullulans* [68,69,70]. These results led to the hypothesis that intracellular levels of glutathione might signal morphological switching in these fungal species. Second, in *H. capsulatum*, the *gsh1* and *gsh2* genes, which encode enzymes responsible for glutathione biosynthesis, were highly expressed in the yeast phase. The overexpression of these genes led to yeast cells being incapable of switching to the mold morphotype, implying that high amounts of glutathione could signal this fungus to remain in the yeast morphotype (Table 1 and Table 2). Third, *Penicillium chrysogenum*, an industrial fungus, underwent yeast-like cell formation in autolysing cultures. During the yeast-like cell transition from the mycelium, there was a relatively high concentration of glutathione and a reductive GSH/GSSH redox balance [71]. The studies from several fungal species consistently suggest that yeast morphotype formation is correlated with high levels of glutathione. A later study in *C. albicans*, however, revealed that glutathione depletion is a consequence of, but not a regulatory signal leading to, the filamentation process [72]. Neither supplementation nor depletion of glutathione affected the ability of *C. albicans* to form hyphae. In fact, glutathione was highly consumed during *C. albicans* filamentation and led to glutathione-dependent oxidative stress. The result from *C. albicans* definitely challenges the role of glutathione as a signaling molecule in fungi, yet also emphasizes that more experiments are needed to unravel the role of glutathione as a regulatory signal in other fungi.

## 6. Concluding Remarks

Glutathione has been called an “altruistic metabolite” in fungi because it participates in the response of cells suffering from oxidative stress [67]. Indeed, the role of glutathione is not limited to stressful conditions. A defective cell in the glutathione metabolic pathway simply cannot survive, even in stress-free situations. Several glutathione genes are considered “essential genes” because the absence of these genes leads to lethality. In pathogenic fungi, glutathione participates in cell differentiation and morphology development, linking glutathione metabolism to virulence attributes. Even though there is no doubt that glutathione is a truly altruistic compound, many questions still remain. For example, it is unclear whether glutathione can function as a regulatory signal to control the morphological switching and virulence pathways in fungi, as seen in the case of bacterial pathogens. How does glutathione metabolism affect cellular function at a global level? How does the cell co-regulate glutathione metabolism and morphogenesis? As transcriptomic and proteomic approaches, as well as CRISPR-based gene editing technology, become more available to the research community, we can study a wider range of genes, proteins and even organisms. We anticipate that with further investigations, more examples will be uncovered to demonstrate how diverse pathogens could use glutathione as a modulator of virulence or how glutathione impacts other cellular functions.

## Figures and Tables

**Figure 1 ijms-23-10645-f001:**
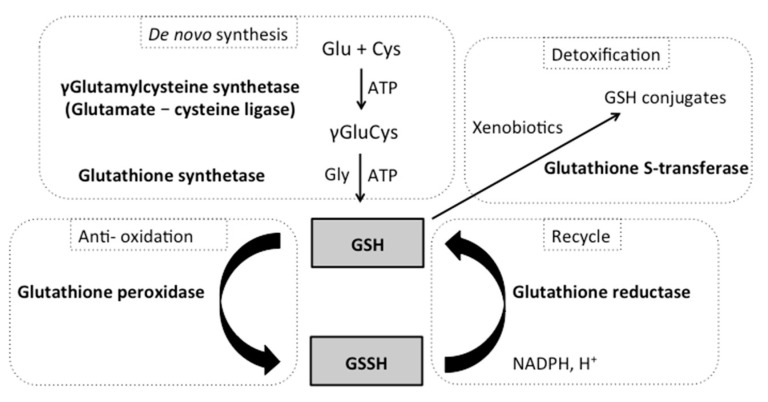
The glutathione pathway in *Saccharomyces cerevisiae*. The synthesis of glutathione (GSH) is a two-step process catalyzed by the enzyme γ-glutamylcysteine synthetase (*GSH1*) and glutathione synthetase (*GSH2*) in the cytoplasm. GSH plays a crucial role in the protection of macromolecules from stressors, especially oxidants. The reduced form (GSH) directly scavenges diverse oxidants and xenobiotics via the action of glutathione peroxidase (*GPX1-3*) and glutathione S-transferase (*GST*) and becomes oxidized (GSSH). Regeneration of oxidized GSSG to reduced GSH is catalyzed by glutathione reductase (*GLR1*) (requires NADPH).

**Table 2 ijms-23-10645-t002:** Role of glutathione levels in fungal cellular development.

Species	Morphological Changes	Role of Glutathione	Reference
*Candida albicans* (Human pathogen)	Yeast-to-hyphae transition	- Intracellular glutathione levels were decreased during a yeast-to-hyphae transition due to high glutathione consumption by filamentous cells. - Glutathione levels were increased in *C. albicans* resistant to fluconazole.	[69,70,72,73,74]
*Aureobasidium pullulans* (Contaminant fungus)	Yeast-to-mycelia transition	Intracellular glutathione levels were higher in yeast cells than in mycelia.	[68]
*Histoplasma capsulatum* (Pathogenic fungus)	Yeast-to-mycelia transition	Glutathione was highly abundant in the yeast form.	[35]
*Coccidioides immitis* (Pathogenic fungus)	Yeast-to-mycelia transition	Genes related to glutathione detoxification pathways were downregulated in yeast spherule form compared to mold form.	[75]
*Penicillium chrysogenum* (Industrial fungus)	Yeast-to-mycelia transition	Intracellular glutathione levels were increased within yeast-like cells in autolysing culture.	[71]
*Saccharomyces cerevisiae* (Model yeast)	Sporulation	- Glutathione was required for sporulation. - Intracellular glutathione levels were decreased during sporulation and completely undetectable during maximum sporulation.	[62,63,76]

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
