# Peer review of "The Role of the Glutathione System in Stress Adaptation, Morphogenesis and Virulence of Pathogenic Fungi"

_ijms, 2022, doi:10.3390/ijms231810645_

Round 1

Reviewer 1 Report

This is a very well written and timely manuscript that summarizes the current status of glutathione research from among a very broad group of pathogenic fungi. It concisely brings together the basic discoveries from a diverse number of studies with regard to fungal glutathione systems to include re-dox reactions as well as the elimination of xenobiotic substances.  Moreover, this review details how the glutathione system plays a role not just in stress adaptation, but also virulence.  The manuscript was easy to read and enjoyable in addition to being very informative.  It describes a potentially unique metabolic pathway to examine with the intent of developing novel chemotherapeutic interventions.  This review should be warmly received by molecular mycologists examining the basis for fungal virulence and host response.

This reviewer has only one significant suggestion to improve the manuscript.  Whereas the text appears to have been written with due care and diligence, the reference section needs to brought to the same level of quality.  In particular, the citations are not necessarily in a consistent style, e.g., some fungal names are italicized, whereas others are not.  A number of citations are noted to be internet derived.  Certainly, these have actual volume numbers, pages, etc.  Some journal titles are presented with the first letter of each word capitalized, but others are not.  In short, this section needs to be addressed.

Finally, I do not believe citation #51 is the relevant reference the authors wish to cite with regard to the Talaromyces marneffei gpx1 gene.  There are a couple of other references that speak more directly to the point trying to be made.

Reviewer 2 Report

This review paper summarises earlier and last publications on glutathione system in fungi having clear focus on functions and expression of glutathione related enzymes (including mutants) and peptide itself. There is no new experimental part of authors. This well written and easy readable review explores the role of glutathione metabolism in fungal adaptation to stress, morphogenesis, and virulence and is relatively comprehensive analysis of fungal glutathione metabolism in and beyond stressful conditions.  Thus, this paper can have larger audience and interest in both molecular biologists and microbiologists.
